# Natural Products in the Treatment of Neuroinflammation at Microglia: Recent Trend and Features

**DOI:** 10.3390/cells14080571

**Published:** 2025-04-10

**Authors:** Chi-Su Yoon

**Affiliations:** College of Pharmacy, Wonkwang University, Iksan 54538, Republic of Korea; ycs1991@naver.com

**Keywords:** natural products, neuroinflammation, microglia, neurodegenerative disease

## Abstract

Natural products (NPs) are considered to be the oldest medicine in human history and numerous NPs have been investigated to search for therapeutic agents in various diseases. Neurodegenerative diseases such as dementia, Parkinson’s, Alzheimer’s, and Huntington’s disease have been increasing following the extension of human lifespans. Neuroinflammation is a key factor in the genesis of several neurodegenerative diseases; therefore, many studies have been focused on finding therapeutics for the reduction in neuroinflammation. Microglia cells are found in the central nervous system (CNS) and these play a crucial role in the regulation of neuroinflammation; thus, the importance of microglia research has been recognized. This review focuses on recent research trends in finding neuroinflammatory regulators in microglia by using NPs.

## 1. Introduction

Incidences of neurodegenerative diseases (ND) such as Parkinson’s disease (PD) and Alzheimer’s disease (AD) have increased dramatically in recent years due to increased life span, the latter of which is a leading cause of mortality [1]. According to an announcement from the Global Dementia Observatory under the World Health Organization (WHO), currently more than 55 million people have dementia worldwide, and every year, there are nearly 10 million new cases. Regardless of the incessant efforts by modern science to create a medical or surgical solution, the outcome has not been favorable. Therefore, NDs remain a clinical concern among the elderly population [2]. They represent a common pathophysiological hallmark of various brain-related disorders [3], and accumulating evidence now demonstrates that neuroinflammation plays a significant role in ND. Neuroinflammation is a protective response of the central nervous system (CNS) to remove harmful stimuli and to initiate a healing process, and microglia are recognized as a playing key role in neuroinflammation process [4]. Microglia are estimated to constitute approximately 5–20% of all cells in the adult human CNS, exhibiting regional heterogeneity in both density and phenotype [5].

Given their critical role, microglial cells have become a major focus of research, particularly in the context of developing therapeutic strategies for ND. Natural products (NPs), derived from a wide range of biological sources including plants, marine organisms, and microbes, have consistently proven to be a rich reservoir of bioactive compounds. These compounds have historically played a pivotal role in pharmaceutical development, offering novel chemical scaffolds and mechanisms of action that have significantly advanced the treatment of numerous human diseases. In recent years, significant research efforts have been directed toward investigating the interactions between NPs and microglial cells, with the aim of identifying potential therapeutic candidates for the treatment of neurodegenerative conditions such as AD, PD, and amyotrophic lateral sclerosis. By targeting microglial activation pathways, these NPs compounds hold promise not only for mitigating the pathological aspects of neuroinflammation but also for enhancing its beneficial effects, such as debris clearance and tissue repair, thereby contributing to the overall management and potential reversal of neurodegenerative processes [6].

This review focused on introducing the general neuroinflammation process in microglia, and in the previously and/or recently discovered NPs which have anti-neuroinflammatory effects on microglia.

## 2. Neuroinflammation in ND

Neuroinflammation plays a crucial role in the CNS, serving as a defense mechanism aimed at eliminating harmful stimuli and promoting tissue repair. This process is primarily mediated by the activation of microglia and astroglia, which work together to maintain CNS homeostasis [7,8,9]. Without an adequate neuroinflammatory response, the CNS would be unable to effectively clear harmful agents or recover from injuries, ultimately resulting in progressive tissue degeneration, functional impairment, and potentially severe consequences [10]. However, when neuroinflammation becomes dysregulated, it can have detrimental effects. To prevent excessive or chronic inflammation, the CNS relies on a highly sophisticated regulatory network to maintain immune balance [4].

In normal conditions, neuroinflammation provides several essential benefits, including pathogen elimination, cellular debris clearance, detoxification, infection containment, and the facilitation of tissue regeneration [11]. At injury sites, activated microglia engage in phagocytosis, engulfing pathogens, dead cells, and other harmful substances. Additionally, microglia secrete cytotoxic molecules, such as reactive oxygen species (ROS), proteases, and pro-inflammatory cytokines, which help neutralize infectious agents and clear damaged neurons. Microglia surrounding senile plaques are stimulated by amyloid peptide to produce interleukin(IL)-1β, that stimulates its own production by binding to its cognate interleukin-1 receptors expressed in microglial cells, further ensuring the amplification of pro-inflammatory signals [12,13].

Beyond eliminating harmful stimuli, neuroinflammation is also crucial for tissue regeneration. Microglia and astroglia coordinate to restore damaged neural structures by secreting anti-inflammatory cytokines, recruiting glial cells to the injury site, and releasing neurotrophic factors. One notable example is glia-derived neurotrophic factor (GDNF), which has been shown to enhance neuronal survival and support the recovery of dopamine neurons in various PD models. Moreover, activated microglia participate in synaptic stripping, a mechanism that removes synaptic inputs from neuronal perikarya near the site of damage, thereby facilitating neural circuit reorganization and regrowth [14].

Furthermore, localized production of chemo-attractants, microglial chemotaxis, and the formation of astrocytic scars help contain tissue damage and prevent infections from spreading [10]. Effective coordination among microglia, astroglia, and neurons is essential to minimize collateral damage to healthy CNS tissue while ensuring an efficient response to injuries and infections. However, in ND, the protective effects of neuroinflammation are often diminished or dysfunctional. A compromised ability to mount a beneficial inflammatory response following CNS injury may undermine the protective role of inflammation, ultimately accelerating the progression of neurodegenerative conditions [13,14].

## 3. Microglial Cells Play a Crucial Role in Neuroinflammation

Since their discovery in 1919, microglia remained poorly understood for decades due to technical limitations. However, recent advancements have revealed their diverse functions and heterogeneity [15]. Traditionally viewed as immunocompetent cells in a quiescent state, microglia are highly sensitive to internal and external stimuli. Upon activation, they shift rapidly, releasing cytokines and chemokines that amplify neuroinflammation (Figure 1) [16]. Beyond immune surveillance, microglia are essential for neuronal maintenance, synapse pruning, and microenvironment regulation, functioning as homeostatic regulators and first responders in the CNS. Beyond their role in immune surveillance, microglia are critical for neuronal maintenance [17]. During brain development, they actively interact with neurons, facilitating synapse pruning, a process essential for neuroplasticity and proper neural circuit formation. Even in the adult brain, microglia remain highly dynamic, constantly extending and retracting their processes to monitor and regulate their microenvironment [18]. Functionally, they operate as homeostatic regulators, immune sentinels, and first responders, ensuring the maintenance of CNS homeostasis and neuronal function [19,20].

The classification of microglial activation states has evolved. Previously categorized as M0 (resting), M1 (pro-inflammatory), and M2 (anti-inflammatory), this framework is now considered oversimplified. Contemporary research suggests a continuum of activation states, ranging from quiescent to fully activated, with intermediate forms identified through gene expression and morphology [21,22]. Microglial heterogeneity across brain regions is now well recognized, with differences in density, morphology, and immune responsiveness influenced by regional localization [17]. For instance, microglia in the cortex and cerebellum display distinct lysosomal degradation capacities. Single-cell RNA sequencing has further revealed specialized subpopulations such as disease-associated microglia (DAM/MGnD) in Alzheimer’s and Parkinson’s models, along with proliferation-associated and lipid-droplet-accumulating microglia, which contribute to both disease progression and neuroprotection [23,24].

Microglia, a subset of glial cells, serve as the primary immune defense mechanism within the CNS. Distributed throughout the mature CNS, they comprise approximately 15% of the total CNS cells and are now widely recognized as originating from mesodermal-derived myeloid progenitors, linking them to the monocyte-macrophage lineage [7]. During embryonic development, these progenitor cells migrate from the yolk sac into the CNS parenchyma, where they establish residency and differentiate into microglia. In adults, the blood-brain barrier (BBB) restricts the infiltration of peripheral immune cells, making it challenging to replace microglia under normal conditions. However, when the BBB is compromised, myeloid progenitors may infiltrate and replenish the microglial population. Unlike peripheral macrophages, microglia exhibit a remarkably low turnover rate while in their resting state. Quiescent microglia are characterized by a ramified morphology, with small, stationary cell bodies and highly motile processes that continuously survey their surroundings. This constant surveillance enables microglia to detect and respond rapidly to subtle changes in the CNS microenvironment, thereby preserving immune homeostasis. Upon injury or immune challenge, microglia become rapidly activated, adopting a phenotype akin to macrophages, but M1 polarized microglia show a decreased capacity for phagocytosis which is instead restored when the cells are pushed towards an anti-inflammatory phenotype [25,26].

While microglia share some functional similarities with peripheral antigen-presenting cells (APCs), such as macrophages, dendritic cells, and B cells, they are considered weak APCs under normal conditions [27]. Effective antigen presentation to T cells requires prior activation, typically occurring under pathological conditions when the BBB becomes compromised. Notably, microglial activation is far more tightly regulated in both space and time compared to peripheral macrophages, ensuring a precise immune response that minimizes collateral damage to vulnerable CNS tissue [28]. In addition to their immunological roles, activated microglia interact closely with astrocytes, facilitating neuroprotection and tissue repair [11]. However, excessive or prolonged microglial activation is not without consequence. The secretion of cytotoxic substances, while initially intended to eliminate infected neurons, bacteria, and viruses, can inadvertently induce bystander damage to surrounding healthy tissue. This underscores the dual nature of the microglial function, where activation serves as both a protective and potentially destructive force within the CNS.

## 4. Signaling Pathway in Neuroinflammation Process

### 4.1. NF-κB Pathway

Nuclear factor-kappa B (NF-κB) is a family of five transcription factors that regulate various cellular processes, particularly those involved in inflammatory responses [29]. This family includes NF-κB1 (p105/p50), NF-κB2 (p100/p52), RelA (p65), RelB, and c-Rel [30]. As a central mediator of inflammation, NF-κB activation induces the transcription of numerous pro-inflammatory genes. The NF-κB signaling pathway is primarily regulated through two distinct mechanisms: the canonical and noncanonical pathways. Among these, the canonical pathway has been extensively studied due to their crucial role in inflammatory responses, which are strongly implicated in the progression of AD. In their inactive state, the p65/p50 dimers of the canonical pathway are retained in the cytoplasm by inhibition kappa B (IκB)α, a key inhibitory protein (Figure 2) [31]. Upon stimulation by pro-inflammatory cytokines, pathogens, or danger-associated molecular patterns (DAMPs), a phosphorylation cascade leads to the proteasomal degradation of IκBα, resulting in the release of the p65/p50 complex. Once freed, this complex translocates into the nucleus, where it binds to κB motifs in the promoter regions of NF-κB target genes, thereby initiating their transcription and promoting inflammatory signaling [3]. In contrast, the noncanonical NF-κB pathway is activated by specific members of the TNF receptor superfamily, leading to the activation of NF-κB-inducing kinase (NIK). Activated NIK phosphorylates IκB kinase alpha (IKKα), subsequently phosphorylates the C-terminal region of p100, leading to their processing into p52. The resulting p52/RelB dimer then translocates into the nucleus, where it drives the expression of genes involved in immune cell development and function [32]. 

### 4.2. Akt/PI3K Pathway

The phosphoinositide 3-kinase (PI3K)/Akt pathway serves as a central regulator of microglial-mediated inflammation, exhibiting dual roles in both promoting inflammation and facilitating neuroprotection [33]. Its activation influences microglial polarization, cytokine production, and overall neuroinflammatory responses, making it a crucial target for therapeutic intervention in neurodegenerative disorders. The PI3K/Akt pathway modulates microglial activation in response to various external stimuli, including lipopolysaccharide (LPS), cytokines, and growth factors. PI3K activation leads to the production of phosphatidylinositol-3,4,5-trisphosphate (PIP3), which recruits and activates Akt, a key kinase involved in regulating inflammation, apoptosis, and cell survival [34]. Studies have shown that inhibiting the PI3K/Akt pathway in LPS-activated microglia significantly reduces the production of pro-inflammatory cytokines, such as tumor necrosis factor-alpha (TNF-α), IL-1β, and IL-6. PI3K/Akt signaling can exert both pro-inflammatory and anti-inflammatory effects depending on the cellular context. While activation of this pathway contributes to inflammation under certain conditions, it also plays a neuroprotective role by upregulating antioxidative responses and inhibiting apoptosis. Specifically, the activation of the PI3K/Akt/mTOR pathway has been associated with neuroprotection, whereas excessive activation may lead to chronic neuroinflammation (Figure 2). Additionally, PI3K/Akt signaling plays a pivotal role in microglial polarization, dictating whether microglia adopt a pro-inflammatory M1 phenotype or an anti-inflammatory M2 phenotype. Increased PI3K/Akt activity has been linked to M2 polarization, which is associated with reduced inflammation and enhanced neuroprotection [35]. Conversely, inhibition of PI3K/Akt has been shown to suppress NF-κB activity, thereby limiting the production of pro-inflammatory mediators. Given its regulatory role in neuroinflammation, the PI3K/Akt pathway represents a promising therapeutic target for mitigating neuroinflammatory responses in ND such as AD, PD, and multiple sclerosis. Several NPs compounds, including curcumin, flavonoids, and polyphenols, have been found to modulate the PI3K/Akt pathway, thereby reducing inflammation and microglial activation. Additionally, pharmacological inhibitors such as LY294002 have been investigated for their ability to block PI3K activity and attenuate microglial-driven neuroinflammation [36,37].

### 4.3. MAPK Pathways

The mitogen-activated protein kinase (MAPK) signaling pathway plays a critical role in regulating microglial inflammatory responses, particularly in ND such as AD. Among the MAPK family members, p38 MAPK and extracellular signal-regulated kinase (ERK) are the most extensively studied in the context of microglial activation. The MAPK pathway modulates the production of pro-inflammatory cytokines, including TNF-α, IL-6, and IL-1β, which are key mediators of neuroinflammation [38,39]. In particular, the p38 MAPK pathway governs TNF-α synthesis, making it a crucial regulator of microglial overactivation and neurotoxicity. In AD models, both the p38 MAPK and ERK pathways are significantly upregulated, correlating with amyloid-beta (Aβ) plaque formation (Figure 2) [40]. Persistent activation of these pathways leads to sustained microglial activation, excessive cytokine release, and neuronal damage, thereby exacerbating disease progression. ERK signaling serves as a fundamental regulator of microglial immune responses, particularly via interferon-gamma (IFNγ) signaling. ERK activation has been linked to the regulation of several disease-associated microglial (DAM) genes, as well as human AD risk genes such as *TREM2*, *BIN1*, and *CD33* [41]. Studies indicate that ERK inhibition suppresses pro-inflammatory gene expression and neuronal phagocytosis, highlighting their potential as a therapeutic target. A key downstream effector of p38 MAPK is MAPK-activated protein kinase 2 (MK2), which modulates the release of TNF-α and IL-8. Notably, deficient microglia in MK2 exhibit reduced inflammatory cytokine production, suggesting that targeting this pathway could attenuate neuroinflammation. Collectively, the MAPK signaling pathway—particularly the p38 MAPK and ERK cascades—plays a pivotal role in regulating microglia-mediated neuroinflammation. Dysregulation of these pathways contributes to the progression of ND by driving chronic inflammation and neuronal damage. Consequently, targeting MAPK signaling may represent a promising therapeutic strategy for mitigating neuroinflammation and preserving neuronal integrity [42]

### 4.4. Nrf2 Pathway

The Nrf2 (Nuclear factor erythroid 2-related factor 2) and Keap1 (Kelch-like ECH-associated protein 1) pathway has been identified as a fundamental mechanism in regulating the antioxidant and anti-inflammatory functions of microglia. Nrf2 is a transcription factor that regulates oxidative stress responses. Under normal conditions, Keap1 mediates Nrf2 degradation in the cytoplasm. Keap1 retains Nrf2 in the cytoplasm and facilitates its ubiquitination under normal conditions, thereby regulating its turnover. Upon oxidative stress, the Keap1 dimer dissociates from Nrf2, allowing Nrf2 to translocate into the nucleus, where it binds to antioxidant response element (ARE) sequences and subsequently promotes the expression of various antioxidant and detoxification genes (Figure 2) [43]. In ND such as AD and PD, microglia become overactivated, releasing inflammatory cytokines and increasing oxidative stress, which leads to neuronal damage. Studies have shown that Nrf2 activation suppresses the expression of pro-inflammatory cytokines (e.g., TNF-α, IL-6) in microglia and promotes the production of antioxidant enzymes (e.g., Superoxide dismutase (SOD), Heme Oxygenase (HO)-1) [44]. Several studies have confirmed that directly inhibiting Keap1 to activate Nrf2 is a promising strategy for treating ND. For instance, using Keap1 inhibitors to enhance Nrf2 nuclear translocation strengthens the antioxidant and anti-inflammatory responses of microglia, thereby providing neuroprotection. Furthermore, certain drugs (e.g., ALGERNON2) have been reported to activate the p21-Nrf2 pathway in microglia, alleviating neurodegeneration caused by neuroinflammation [45]. As a conclusion, the Nrf2-Keap1 pathway in microglia plays a critical role in regulating neuroinflammation and mitigating oxidative stress. Targeting this pathway presents a novel therapeutic strategy for ND. In particular, activating Nrf2 through Keap1 inhibition effectively modulates inflammatory responses and protects neurons, making it a key mechanism in slowing the progression of neurodegenerative disorders [46,47].

### 4.5. CREB Signaling

The cAMP response element-binding protein (CREB) signaling pathway plays a crucial role in microglial function, particularly in response to Aβ-induced neurotoxicity in AD. CREB is a transcription factor essential for synaptic plasticity, memory formation, and neuronal survival, but its dysregulation in microglia can contribute to neurodegeneration. The Aβ42 infusion in mice led to a transient increase in CREB phosphorylation (pS133-CREB) in microglia, correlating with memory impairment and inflammatory responses. Interestingly, while CREB phosphorylation decreased in neurons, it was upregulated in activated microglia, suggesting a cell-type-specific response. Elevated pS133-CREB in microglia was associated with increased levels in pro-inflammatory cytokines (IL-6) and enhanced binding of CREB to matrix metalloproteinase-9 (MMP-9) DNA, further amplifying neuroinflammation. Blocking microglial activation with minocycline or inhibiting CREB phosphorylation via the PKA inhibitor H89 significantly attenuated Aβ-induced neurotoxicity, restoring neuronal function and synaptic protein expression. Additionally, co-culturing hippocampal neurons with Aβ-stimulated microglial media reduced synaptic protein levels (GluN1, GluA2), while PKA inhibition reversed these effects, indicating that microglial CREB activation contributes to Aβ-induced neuronal damage. The CREB signaling in microglia plays a dual role while it regulates neuroprotection and homeostasis under normal conditions, Aβ-induced CREB hyperactivation leads to inflammatory responses and neuronal dysfunction. Modulating CREB phosphorylation in microglia may represent a potential therapeutic target for reducing neuroinflammation and preserving neuronal integrity in AD [48,49].

### 4.6. JAK/STATs

The Janus kinase/signal transducer and activator of transcription (JAK/STAT) pathway is a crucial signaling cascade involved in neuroinflammation and ND. Microglia utilize this pathway to regulate immune responses, cell survival, and neurotoxic activities. The JAK/STAT pathway plays a pivotal role in this activation process [50]. STAT1, a key component of the pathway, promotes M1 polarization, leading to the production of inflammatory cytokines such as IFN-γ, IL-6, TNF-α, and ROS, which can exacerbate neuroinflammation. Conversely, STAT3 has a more dual role, contributing to both neuroprotection and neurotoxicity depending on the context [51]. Dysregulation of JAK/STAT signaling in microglia has been linked to neurodegenerative disorders such as AD and PD. In AD, chronic activation of the pathway exacerbates neuroinflammation through sustained microglial activation, leading to increased β-amyloid aggregation and neuronal damage. Similarly, in PD, JAK/STAT-mediated microglial activation enhances synuclein toxicity, further promoting neurodegeneration. Recent studies highlight the role of hypoxia in microglial activation through STAT1. Under low oxygen conditions, STAT1 becomes aberrantly activated via oxidative stress-induced modifications, such as *S*-glutathionylation, leading to persistent M1 microglial activation and neurotoxicity. This suggests that targeting STAT1 could be a potential therapeutic strategy to mitigate hypoxia-induced neuroinflammation. Given their critical role in neuroinflammation, the JAK/STAT pathway has become a promising target for therapeutic interventions. Several JAK inhibitors are being explored to modulate excessive microglial activation and reduce neuroinflammatory damage. By carefully regulating this pathway, it may be possible to shift microglial activity toward a neuroprotective state, providing new avenues for treating ND. In conclusion, the JAK/STAT pathway is essential in regulating microglial function and neuroinflammation. Their dysregulation contributes to the progression of neurodegenerative disorders, particularly through STAT1-mediated pro-inflammatory responses. Targeting this pathway holds potential for therapeutic strategies aimed at reducing neurotoxicity and promoting neuroprotection in CNS diseases [52,53,54].

## 5. Potent NP Candidates for Regulation of Neuroinflammation to Treat ND

### 5.1. Previous NP-Derived Compounds for Treat ND

Following a review paper, a number of NPs were identified as having bioactivity relevant to a specific ND. Among the 537 compounds studied, 71.9% were associated with AD, 2.4% with PD, 2.2% with Huntington’s disease (HD), 1.7% with prion diseases, and 21.8% were related to general neurodegeneration. The origin of NPs with bioactivity relevant to neurodegeneration is shown as a percentage of the total identified compounds (*n* = 204). The majority originate from plants (72.1%), followed by fungi (12.7%), marine sources (9.3%), bacteria (3.4%), insects (2.0%), and algae (0.5%). NPs are expected to play a crucial role in developing new therapeutic leads for ND [55]. Secondary metabolites have historically been a valuable source for small-molecule therapeutics, partly because of their unique chemical properties and strong bioactivities. Previous studies have highlighted specific subsets of NPs that show potential for treating AD and PD, and some of the NPs-derived molecules for treatment were reviewed in a clinical study for the treatment of ND [56] (Table 1, Figure 3).

Table 2 and Figure 3 indicate that various types of natural product-derived compounds have been investigated for their potential to inhibit neuroinflammation in microglial cell lines. This review summarized several recent studies that have utilized different experimental models by using well-known compounds.

1,6-*O*,*O*-Diacetylbritannilactone (OABL) is a sesquiterpene lactone isolated from the *Inula* species. The genus *Inula*, widely distributed across Asia and Europe, has traditionally been used in herbal medicine for the treatment of bronchitis, diabetes, intestinal ulcers, digestive disorders, and inflammation [57]. In a study by Wang et al. [58], OABL demonstrated promising anti-neuroinflammatory activity, along with favorable BBB permeability. Their findings revealed that OABL inhibits the NF-κB transcription factor, and site-specific profiling identified NLRP3 as a covalent target through quantitative thiol reactivity profiling (QTRP). In a separate study by Tang et al. [59], OABL attenuated LPS-induced neuroinflammation in BV-2 microglial cells by suppressing inflammatory mediator levels. It also exhibited neuroprotective effects against oxytosis and ferroptosis in the rat pheochromocytoma PC12 cell line. In vivo, OABL improved cognitive function in 6-month-old 5xFAD mice and significantly reduced amyloid plaque accumulation, Aβ expression, Tau phosphorylation, and BACE1 expression in the brains of AD model mice.

Magnolol and honokiol, two neolignan compounds found in *Magnolia officinalis*, are widely used as dietary supplements and are known for their potent antioxidant and anti-inflammatory properties, and magnolol is considered a safe compound [60]. In a study by Tao et al. [61], magnolol exhibited antidepressant effects by inhibiting inflammation, suppressing pro-inflammatory cytokines, promoting anti-inflammatory cytokines, and enhancing the transcription of M2 microglial phenotype markers. Additionally, it upregulated Nrf2 and HO-1 expression, while downregulating NLRP3, caspase-1 p20, and IL-1β expression both in vivo and in vitro. Similarly, Xie et al. [62] reported that magnolol inhibited NF-κB luciferase activity and the expression of its downstream pro-inflammatory cytokines—a process that was blocked by GW9662, a PPAR-γ antagonist. Furthermore, magnolol activated the Nrf2-ARE pathway and reduced Aβ-induced ROS levels.

Liquiritigenin and isoliquiritigenin are flavonoid constituents of *Glycyrrhiza glabra* (licorice). While structurally similar, liquiritigenin possesses a flavanone skeleton, and isoliquiritigenin contains a chalcone backbone. Both compounds have shown anti-neuroinflammatory effects in various studies. In a study by Du et al. [63], liquiritigenin reduced Aβ levels and ameliorated cognitive decline in AD mice by modulating microglial M1/M2 polarization. Bai et al. [64] demonstrated that isoliquiritigenin inhibited neuroinflammation in both a 1-methyl-4-phenylpyridinium (MPTP)-induced PD mouse model and in LPS-stimulated BV-2 microglial cells. These effects were associated with inhibition of the JNK, AKT, and NF-κB signaling pathways. Similarly, Lee et al. [65] showed that isoliquiritigenin attenuated inflammation in LPS-induced BV-2 microglial cells by suppressing ERK/p38/NF-κB activation and ROS generation. Interestingly, their study also suggested a link between these effects and mitochondrial fission as well as the calcium/calcineurin signaling pathway. In a different context, Wang et al. [66] reported that isoliquiritigenin alleviated neuropathic pain by reducing microglial inflammation through ERK pathway inhibition and downregulation of CEBPB transcription. The analgesic effect of isoliquiritigenin was primarily attributed to its suppression of spinal microglial activation and neuroinflammation.

Sulforaphane, an isothiocyanate type compound, is a known component in Cruciferous vegetables such as sprouts, broccoli, radish cabbage, and wasabi [67]. In a study by Yang et al. [68], sulforaphane inhibited cytostatic autophagy-induced activation of the NLRP3 signaling pathway in Aβ-stimulated microglia by reducing ROS production, thereby attenuating neuronal damage. Similarly, Qin et al. [69] demonstrated that sulforaphane mitigated neuronal necroptosis by downregulating the MAPK/NF-κB signaling pathways in LPS-activated BV-2 microglial cells. These findings provide new insights into the therapeutic potential of sulforaphane in AD.

Ginseng, a perennial herb, has long been used in various countries for its medicinal properties. In traditional Chinese medicine, it is known as the “king of herbs” due to its reputed anti-aging, anti-inflammatory, and anti-apoptotic effects [70]. Ginsenosides, which are triterpenoid saponins, constitute the major active components of ginseng. Numerous types of ginsenosides have been studied for their diverse biological functions. In a study [71], ginsenoside Ro was shown to ameliorate neuroinflammation by reducing the presence of IBA1-positive microglia and GFAP-positive astrocytes, decreasing pro-inflammatory cytokines, and increasing anti-inflammatory cytokines. In a study by Liu et al. [72], ginsenoside Rg3 alleviated neuroinflammation in primary microglia and hippocampal neuronal damage in a traumatic brain injury (TBI) mouse model. The compound exerted its protective effects through modulation of the SIRT1/NF-κB pathway, suggesting its therapeutic potential for TBI. Madhi et al. [73] reported the anti-neuroinflammatory effects of ginsenoside Re. This compound inhibited the production of inflammatory mediators in BV-2 microglial cells and nitric oxide in primary microglia. Furthermore, ginsenoside Re suppressed the activation of NF-κB, Ca^2+^/calmodulin-dependent protein kinase (CaMK)2 and CaMK4, ERK, and JNK pathways. It also conferred protective effects in hippocampal neurons by reducing the release of inflammatory and neurotoxic factors from activated microglia.

This section has summarized recent studies on NPs-derived single compounds exhibiting anti-neuroinflammatory effects in microglial cells. In the following section, recent investigations of NPs compounds or extracts in novel neurodegenerative disease models will be discussed.

**Table 2 cells-14-00571-t002:** Recent studies on different types of natural products inhibiting neuroinflammation in microglia cells.

Category	Natural Products	Source	Model	Biological Target	Exp. Dose Range	Ref.
Sesquiterpene	1,6-*O*,*O*-diacetylbritannilactone (OABL)	*Inula * *japonica*	BV2 cell (in vitro) 5xFAD mice (in vivo)	Inhibition of NF-κB pathways, binding to NLRP3	10 μM 20 mg/kg	[58]
			BV2 cell (in vitro) 5xFAD mice (in vivo)	Inhibition of NF-κB pathways, reduces Aβ accumulation and p-Tau level	10 μM 20 mg/kg	[59]
Lignan	Magnolol	*Magnolia officinalis*	Iba-1+CD16/32+, Iba-1+CD206+, BV2 cell (in vitro), depression model mice (in vivo)	Inhibiting M1 polarization and inducing M2 polarization via Nrf2/HO-1/NLRP3 signaling	100 mg/kg	[61]
			*C. elegans* (in vivo), BV2 cell (in vitro)	Promoting microglia phagocytosis and the degradation of beta-amyloid through activation of PPAR-γ	10 μM	[62]
Flavonoid	Liquiritigenin	*Glycyrrhiza glabra* (licorice)	AD mice (in vivo)	Decreases Aβ levels and ameliorates cognitive decline by regulating microglia M1/M2 transformation	30 mg/kg	[63]
Chalcone	Isoliquiritigenin		BV2 cell (in vitro), PD mice (in vivo)	Inhibition of JNK/AKT/NFκB signaling pathway	40 μM 40 mg/kg	[64]
			BV2 cell (in vitro)	Inhibition of ERK/p38/NF-κB activation and preventing mitochondrial fission	10 μM	[65]
			BV2 cell (in vitro) CCI mouse model (in vivo)	Inhibition of CEBPB/ERK pathway, reducing neuropathic pain in mice	80 μM 40 mg/kg	[66]
Isothiocyanate	Sulforaphane	Cruciferous vegetables	primary microglia (in vitro)	Inhibition of ROS/autophagy/NLRP3 signal axis	10 μM	[68]
			BV2 cell (in vitro)	Inhibition of MAPK/NF-κB signaling pathways	10 μM	[69]
Triterpenoid (ginsenoside)	Ginsenoside Ro	*Panax ginseng* (ginseng)	APP/PS1 transgenic mice (in vivo)	Inhibition of IBA1/GFAP-MAPK signaling pathway	15 mg/kg	[71]
	Ginsenoside Rg3		TBI mice (in vivo), primary microglia (in vitro)	Inhibition of NF-kB pathway via SIRT1 activation	10 mg/kg, 20 μM	[72]
	Ginsenoside Rh2		Offspring mice (in vivo) from maternal toxoplasma infection during pregnancy	Inhibition of HMGB1/TLR4/NF-kB signaling pathway	100 mg/kg	[73]

### 5.2. The New Features of Recent NP Studies by Regulating Microglia Associated Neuroinflammation

#### 5.2.1. Molecular Docking Simulation

Computational methods for understanding ligand or protein interactions are among the fastest approaches for identifying potential drug candidates and targets. As a strategy, scientists are screening molecules to find compounds that may be effective.

Liu et al. researched on the anti-neuroinflammatory effects of prenylated indole alkaloids from the Antarctic fungus *Aspergillus* sp. strain SF-7367. In the study, metabolites from the fungal strain revealed five known compounds: epideoxybrevianamide E, brevianamide V/W, K, Q, and R, and among these compounds, brevianamide K showed significant anti-inflammatory activity through inhibition of NF-κB signaling. A molecular docking study was used to predict the interaction between the other four compounds, and they found that brevianamide K interacts with p65 protein in NF-κB signaling pathway [74].

*Eleutherococcus henryi* (EH), known as “Wu-Jia-Pi” in traditional Chinese medicine, has been historically used for conditions such as amnesia, mental fatigue, arthritis, and rheumatism. Despite its long-standing medicinal use, scientific research on its neuroprotective effects remains limited. This study aimed to explore the anti-neuroinflammatory properties of EH and identify its bioactive compounds. Using chromatographic techniques, 31 phytochemicals were isolated from the methanol extract of EH root bark, including seven new compounds and one novel natural product. The ethyl acetate fraction of EH methanol extract (EHME) significantly inhibited NO release in LPS-activated BV2 microglia, indicating anti-neuroinflammatory activity. In vitro assays identified multiple active compounds (lignan and phenolic derivates). Among the compounds, the lignan type compound, eleuhenryiside I, showing the highest efficacy. Network pharmacology, molecular docking, and molecular dynamics simulations predicted that eleuhenryiside I targets key neuroinflammatory pathways, including TLR4, Src, MAPK, and NF-κB. In vitro validation confirmed that eleuhenryiside I exerts its anti-neuroinflammatory effects by modulating the TLR4/Src/MAPK p38/NF-κB p65 signaling pathway. This study is the first to identify EH-derived bioactive compounds with significant neuroprotective potential. The findings provide scientific support for the traditional use of EH in neurological disorders and suggest that its lignan compounds, particularly eleuhenryiside I, could serve as promising candidates for anti-neuroinflammatory drug development [75].

Particulate matter (PM2.5) containing polycyclic aromatic hydrocarbons (PAHs) has adverse effects on human health, including ND. *Areca catechu* L. (AC) fruit possesses various pharmacological properties, but its anti-neuroinflammatory roles against PAH-induced neuroinflammation are limited. This study examines AC’s effects and related signaling cascades in anthracene-induced toxicity and inflammation in mouse microglial BV-2 cells. LC-MS identified significant bioactive compounds like arecoline, (−)-epicatechin, and syringic acid in the ethanolic extract of AC (ACEE). Molecular docking revealed (−)-epicatechin had the highest binding affinities against NF-κB. Anthracene induced intracellular ROS, TNF-α, IL-1β, and IL-6 mRNA levels, and TNF-α release via JNK, p38, and NF-κB pathways. ACEE or (−)-epicatechin co-treatment reversed these changes, suggesting potential as anti-neuroinflammatory agents for preventing inflammation-mediated NDDs. However, further research on ACEE and its components in other glial cells, animal models, and clinical trials is needed to fully understand their effects and therapeutic potential. Despite the carcinogenic classification of the areca nut due to arecoline, other compounds within it may provide beneficial effects on neuroinflammation without severe adverse effects. This study highlights the potential anti-neuroinflammatory properties of ACEE, which can be further developed as agents for treating air pollution-induced neuroinflammation in the CNS [76].

Another study explored the anti-aging effects of *Taxus chinensis* fruit extract (TCFE), traditionally used in China for its health benefits. Using a D-galactose-induced aging mouse model, researchers administered TCFE at different doses and conducted behavioral, biochemical, and histological analyses, along with in vitro studies on LPS-stimulated BV2 microglial cells. TCFE improved cognitive function, reduced oxidative stress markers (MDA), increased antioxidants (SOD, T-AOC), and lowered pro-inflammatory cytokines (IL-1β, IL-6, IFN-γ, TNF-α, IL-17). It also inhibited microglial activation and suppressed the TLR4/NF-κB/NLRP3 inflammatory pathway, surpassing the effects of rapamycin and metformin. UPLC-MS/MS analysis identified ten bioactive compounds, including gallocatechin, epigallocatechin, catechin, and quercetin, all showing strong TLR4 binding affinity. In vitro, TCFE reduced IL-1β, NF-κB, and TLR4 levels, similar to the TLR4 inhibitor C34. These findings suggest that TCFE modulates oxidative stress and inflammation, making it a promising candidate for anti-aging therapies and future drug development [77].

#### 5.2.2. Network Pharmacology

Network pharmacology plays a crucial role in natural product research, especially in traditional Chinese medicine (TCM), by analyzing multi-target and multi-component interactions. Unlike the conventional “one drug–one target–one disease” model, network pharmacology enables a systematic understanding of complex biological mechanisms. It can play a key role in NPs research, such as mechanism elucidation; target prediction and drug discovery; multi-component, multi-target analysis; and precision medicine applications [78].

*Tinospora sinensis* (TIS) is valued in Tibetan medicine for treating rheumatic and aging diseases. This study explores its anti-neuroinflammation efficacy, major active ingredients, and mechanisms. Using UPLC-Q-TOF/MS, 39 compounds were identified in TIS, including genipingentiobioside, isocorydin, reticuline, (−)-argemonine, tinosineside A, tinosinenside A, and costunolide. A neuroinflammation model was established by injecting C57BL/6 J mice with LPS. TIS improved learning and memory abilities, reduced inflammatory factors (IL-1β, TNF-α, IL-6, iNOS), and decreased microglial activation in the brain. Network pharmacology and proteomics predicted TNF, IL-1β, IκBKB, and identified key proteins (TNF, NF-κB2, NF-κBIA, TLR4) in neuroinflammation pathways. TIS alleviated neuroinflammation by inhibiting the TLR4/NF-κB/NLRP3 pathway, showcasing its potential as a treatment for neuroinflammatory conditions in AD [79].

In another study, the neuroinflammatory effects of *Artemisia argyi* using network pharmacology and experimental validation were explored. Researchers isolated 17 sesquiterpenoids, including two newly identified compounds, argyinolide S (AS) and argyinolide T (AT). Among them, AS exhibited the strongest anti-inflammatory activity and was selected for further investigation. Using network pharmacology tools such as PharmMapper and TargetNet, key targets of AS were predicted, revealing 11 core proteins, including AKT1 and epidermal growth factor. Molecular docking and dynamics simulations indicated a strong binding affinity between AS and JAK1, suggesting its role in modulating the JAK1/STAT3 signaling pathway. These predictions were validated through drug affinity responsive target stability (DARTS) and Western blot analyses, confirming AS’s inhibitory effect on JAK1, leading to reduced neuroinflammation in BV-2 microglial cells. Notably, AS demonstrated superior efficacy to dexamethasone, highlighting its potential as a novel neuroinflammation inhibitor. The study provides a theoretical foundation for the therapeutic application of *A. argyi* sesquiterpenoids in treating neuroinflammatory diseases and supports the integration of network pharmacology in natural product-based drug discovery [80].

## 6. Conclusions and Perspective

At present, there are no fully effective treatments available for AD, PD, HD, and prion diseases. However, the extensive range of promising secondary metabolites discussed in this review highlights the potential of NPs as a valuable resource for novel drug discovery. Given the diversity of bioactive compounds derived from natural sources, screening programs focused on evaluating these molecules are likely to identify new drug candidates capable of targeting ND pathways through innovative mechanisms. This multi-target approach is expected to offer greater therapeutic value in addressing the complexity of neurodegeneration. The pharmacological advantages of NPs, including their safety and efficacy, position them as promising alternatives to conventional treatments. Evidence from previous research suggests that NPs targeting anti-inflammatory and downstream signaling pathways have shown promising results in both in vivo and in vitro studies, indicating their potential to delay or prevent AD progression. This review provides a comprehensive discussion on the anti-neuroinflammatory in microglia cells, outlining neuroinflammation signaling pathways and emphasizing the NP role in finding ND treatment. Given their anti-inflammatory, antioxidant, and anticholinesterase properties, NPs represent a promising class of therapeutic agents for ND. In response to global population aging, there is an urgent need for increased efforts in developing products based on the advantages of NPs, as well as for conducting more extensive clinical research. Furthermore, advances in artificial intelligence (AI) have enabled the identification of direct interactions between receptors and compounds in brain molecular models. Accordingly, more research is needed to predict structure–activity relationships (SAR) of known bioactive natural compounds in order to discover more stable and potent derivatives. Moreover, alongside AI-driven predictive studies, it is essential to verify whether the predicted outcomes are consistently reproduced in experimental models, thereby ensuring the reliability and translational potential of AI-based approaches.

## Figures and Tables

**Figure 1 cells-14-00571-f001:**
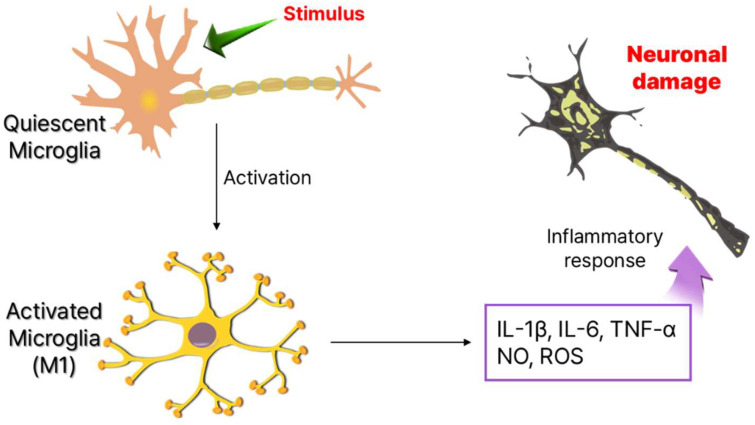
Phenotype of microglia cells and neuronal damage by production of inflammatory cytokines.

**Figure 2 cells-14-00571-f002:**
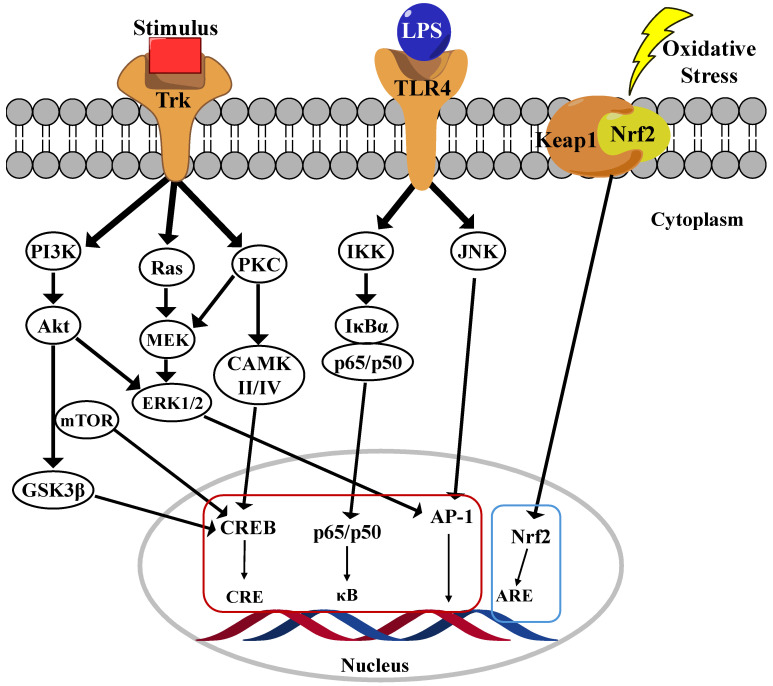
Main neuroinflammation signaling pathway in microglia cells.

**Figure 3 cells-14-00571-f003:**
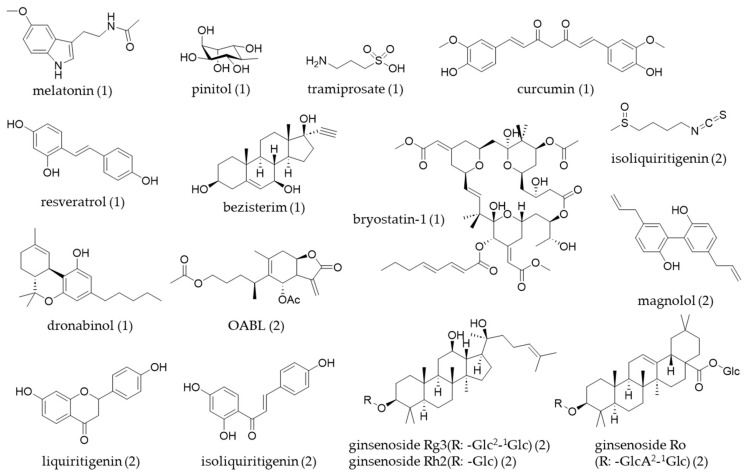
The chemical structures of natural products (table number) covered in this review.

**Table 1 cells-14-00571-t001:** NPs-derived molecules for treatment of ND in clinical trial (2008–2023).

Compound	Origin	Therapeutical Target for ND	Trial Number (Authority, Phase)
Melatonin	Bacteria, eukaryotes	NLRP3, NF-κB pathway	NCT00940589 (FDA, 2)
Pinitol	Soybenans and fruits	Amyloid beta 42	NCT00470418 (FDA, 2)
Tramiprosate	Seaweed	Amyloid beta 42	NCT00314912 (FDA, 3)
Resveratrol	Grapes, peanuts	Multiple pathways	NCT01504854 (FDA, 2)
			NCT00678431 (FDA, 3)
Curcumin	Curcuma	Multiple pathways	NCT00099710 (FDA, 2)
Bezisterim	Adrenal sterol metabolite	ERK 1/2, NF-κB pathway	NCT04669028 (FDA, 3)
			NCT05227820 (FDA, 2)
Bryostatin-1	Bryzoan	Protein kinase C pathway	NCT04538066 (FDA, 2)
Dronabinol	Cannabis	CB1/CB2 receptor agonist	NCT02792257 (FDA, 2)

## Data Availability

No new data were created or analyzed in this study.

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
