# Peer review of "Natural Products in the Treatment of Neuroinflammation at Microglia: Recent Trend and Features"

_cells, 2025, doi:10.3390/cells14080571_

Round 1

Reviewer 1 Report

Comments and Suggestions for Authors

The review entitled “Natural products in the treatment of neuroinflammation at microglia: recent trend and features” by   Chi-Su Yoon describes in detail the role of microglia in the development of neuroinflammation that underlies the main neurodegenerative diseases. The main pathways involved in inflammatory processes at the CNS level are extensively analyzed and some bioactive compounds obtained from extracts of natural components with anti-inflammatory and anti-aging activity are listed. Furthermore, the paper introduces innovative methods, such as network pharmacology, molecular docking, and molecular dynamics simulations: to predict the anti-inflammatory properties of molecules obtained from plant extracts. The work is interesting and well presented.

Below are some comments:

References are missing. Please Insert a bibliographical entry at the end of each sentence that reports the results of previous studies. The lack of citations is confirmed by the number of citations currently present in the reference list: only 38 entries when usually a review exceeds 100 entries.

Lines 68-69 : “microglia are frequently observed clustering around extracellular senile plaques, actively degrading and removing amyloid deposits” Microglial cells have been detected around amyloid plaques, but the interpretation given in previous studies has been different Microglia surrounding senile plaques are stimulate by amyloid peptide to produce IL-1 beta , that stimulates its own production by binding to its cognate interleukin-1 receptors expressed in microglial cells, further ensuring the amplification of pro-inflammatory signals  (Rothwell JN and L. GN (2000). Interleukin 1 in the brain: biology, pathology and therapeutic target Trends in Neuroscience 23, 618–625.; Araujo DM, Cotman CW. Beta-amyloid stimulates glial cells in vitro to produce growth factors that accumulate in senile plaques in Alzheimer's disease. Brain Res. 1992;569:141-5)

Lines 131-32 “Upon injury or immune challenge, microglia become rapidly acti-131 vated, adopting a phenotype akin to macrophages, characterized by phagocytosis”.  M1 polarized microglia shows a decreased capacity for phagocytosis which is instead restored when the cells are pushed towards an anti-inflammatory phenotype

Line 236: “Keap1 inhibits Nrf2 degradation, allowing Nrf2 to translocate into the” . Keap1 keeps Nrf2 in the cytoplasm and provides for its ubiquitination for normal turnover. Oxidative stress dissociates the Keap1 dimer from Nrf2, allowing its translocation into the nucleus, amd the subsequent binding to ARE sequences

Minor:

Please insert the abbreviation for natural products (NP) in the abstract

Line 26 please reconsider the sentence: something is not correct

Line 159, please change its in their

Line 242 ..”thereby exerting…” the sentence is truncated

Line 257 the abbreviation AD has benn already entered: there is no need to indicate it here again

Line 278: please delete “resident immune cells of the CNS”, it is a repetition of what was said before

Comments on the Quality of English Language

the sentences are clear but sometimes there are errors in the verb forms; there are typing errors

Author Response

Comment 1. References are missing. Please Insert a bibliographical entry at the end of each sentence that reports the results of previous studies. The lack of citations is confirmed by the number of citations currently present in the reference list: only 38 entries when usually a review exceeds 100 entries.

Answer: Thank you for your meaningful comments. I tried to add many of the original references in the manuscript.

Comment 2. Lines 68-69 : “microglia are frequently observed clustering around extracellular senile plaques, actively degrading and removing amyloid deposits” Microglial cells have been detected around amyloid plaques, but the interpretation given in previous studies has been different Microglia surrounding senile plaques are stimulate by amyloid peptide to produce IL-1 beta , that stimulates its own production by binding to its cognate interleukin-1 receptors expressed in microglial cells, further ensuring the amplification of pro-inflammatory signals  (Rothwell JN and L. GN (2000). Interleukin 1 in the brain: biology, pathology and therapeutic target Trends in Neuroscience 23, 618–625.; Araujo DM, Cotman CW. Beta-amyloid stimulates glial cells in vitro to produce growth factors that accumulate in senile plaques in Alzheimer's disease. Brain Res. 1992;569:141-5)

Answer: I sincerely appreciate your insightful suggestions. The manuscript has been revised in accordance with your suggestions.

Comment 3. Lines 131-32 “Upon injury or immune challenge, microglia become rapidly acti-131 vated, adopting a phenotype akin to macrophages, characterized by phagocytosis” M1 polarized microglia shows a decreased capacity for phagocytosis which is instead restored when the cells are pushed towards an anti-inflammatory phenotype

Answer: I greatly appreciate your constructive feedback. According to your suggestion, I revised the sentence.

Comment 4. Line 236: “Keap1 inhibits Nrf2 degradation, allowing Nrf2 to translocate into the” . Keap1 keeps Nrf2 in the cytoplasm and provides for its ubiquitination for normal turnover. Oxidative stress dissociates the Keap1 dimer from Nrf2, allowing its translocation into the nucleus, amd the subsequent binding to ARE sequences Neurochem Int. 1997, 30, 433-9

Answer: Thank you for your meaningful comments. I revised the manuscript following your suggested comments.

Comment 5. Please insert the abbreviation for natural products (NP) in the abstract

Answer: Thank you for your meaningful comments. I revised all the abbreviations regarding natural products (NP).

Comment 6. Line 26 please reconsider the sentence: something is not correct

Answer: Thank you for your meaningful comments. I edited the sentence as you pointed out.

Comment 7. Line 159, please change its in their

Answer: Thank you for your meaningful comments. I edited the manuscript.

Comment 8. Line 242 ..”thereby exerting…” the sentence is truncated

Answer: Thank you for your meaningful comments. I revised the sentence.

Comment 9. Line 257 the abbreviation AD has benn already entered: there is no need to indicate it here again

Answer: Thank you for your valuable comments. I have revised the abbreviations related to AD accordingly.

Comment 10. Line 278: please delete “resident immune cells of the CNS”, it is a repetition of what was said before

Answer: Thank you for your meaningful comments. I deleted the sentence according to your suggestion.

Reviewer 2 Report

Comments and Suggestions for Authors

The review article entitled "Natural products in the treatment of neuroinflammation at microglia: recent trend and features"  summarys the Neuro-inflammation in neurodegenerative diseases, such Parkinson's disease, Alzheimer's Disease, Huntinton's disease, and the potential role of microglia, as well as the signaling pathway. Finally, both molecular docking simulation and network pharmacology are introduced. The topic is interesting, however,the Prospects part is too brief. The subheading should be "Conclusion and perspective". Also, The overall figure quality should be further improved.

Comments on the Quality of English Language

The manuscript is suggested to be revised by a native English speaker or editor.

Author Response

The review article entitled "Natural products in the treatment of neuroinflammation at microglia: recent trend and features"  summarys the Neuro-inflammation in neurodegenerative diseases, such Parkinson's disease, Alzheimer's Disease, Huntinton's disease, and the potential role of microglia, as well as the signaling pathway. Finally, both molecular docking simulation and network pharmacology are introduced.

Comment 1. The topic is interesting, however,the Prospects part is too brief. The subheading should be "Conclusion and perspective". Also, The overall figure quality should be further improved.

Answer: Thank you for your valuable comments. I have added a discussion on perspective in the conclusion and modified the section's subject accordingly. Additionally, I have revised the manuscript by incorporating more examples of natural products. Please refer to the updated version of the manuscript.

Reviewer 3 Report

Comments and Suggestions for Authors

Dear Authors:

The submitted manuscript (cells-3530799) entitled “Natural products in the treatment of neuroinflammation at microglia: recent trend and features” discussed the role of NPs in regulating neuroinflammation, particularly through their effects on microglia cells, highlighting their potential as multi-target therapies. However, despite the importance of this topic, this manuscript requires significant revisions to enhance its focus and depth before it is suitable for publication in Cells as follows:

  1. According to the Manuscript Title and Introduction (Lines 50–51), a connection between NPs and the treatment of neuroinflammation at the microglial cell level should be focused, clarified, and explained in detail. However, the authors have overlooked numerous citations and an in-depth discussion on how microglial cells modulate the neuroinflammation through the effects of various NPs (e.g., alkaloids, flavonoids, etc.) on their membranes or specific targets, as illustrated in Figure 2. Examples of relevant citations are provided in the reference list below, and the authors are encouraged to include more references to strengthen this perspective. Additionally, the authors should compile this information into a detailed new table with the following column headings:
    • Natural product (Extract or Compound name)
    • Mechanism of Action (Linked to microglial cell targets)
    • Model used (e.g., In vitro, In vivo, molecular modeling, with or without molecular dynamics simulations [MDS], etc.)
    • Cited references
  2. Based on the expanded discussion and the new table, as suggested above, Figure 2 should be revised to reflect the updated information.
  3. Detailed information on active compounds, e.g., those mentioned in Line 343: "In vitro assays identified multiple 342 active compounds (4–17, 19, 20, 22, 23, 26, 29, and 31), with compound 6 showing..." are required. Each compound should be individually named as mentioned in literature, its chemical nature, along with the corresponding reference cited. Additionally, the chemical identity/name of "compound 6" and other numbered compounds throughout the manuscript must be explicitly provided.
  4. All abbreviations must be defined upon their first appearance in the text. Examples include RelA, IκB, c-Rel, TREM2, PKa, etc., which currently lack full explanations for the reader.
  5. For Table 1, which lists NPs treating neurological diseases (irrespective of their effects on microglial cells), additional columns should be added for: cited references and clinical trial phase (I, II, III, in market).
  6. Latin names of plants or organisms should be italicized throughout the manuscript for consistency and accuracy.

References

Examples of NPS acting on microglia/microglial targets: resveratrol, cannabidiol, ginsenosides, flavonoids and sulforaphane (Maurya et al. 2021) – NP-derived compounds pounds (Park et al. 2023) - tunicatachalcone (Wen et al. 2022) - alpha-Asarone (Cai et al. 2016) - gonadal steroid hormones (Johann and Beyer 2013) - isoquinoline alkaloids - flavonoid rutin and its aglycone quercetin (da Silva et al. 2020) – and more …………………………………………….

References:

  • Cai Q, Li Y, Mao J, Pei G. 2016. Neurogenesis-Promoting Natural Product alpha-Asarone Modulates Morphological Dynamics of Activated Microglia. Front Cell Neurosci.10:280. Epub 2016/12/27.
  • da Silva AB, Cerqueira Coelho PL, das Neves Oliveira M, Oliveira JL, Oliveira Amparo JA, da Silva KC, Soares JRP, Pitanga BPS, Dos Santos Souza C, de Faria Lopes GP, et al. 2020. The flavonoid rutin and its aglycone quercetin modulate the microglia inflammatory profile improving antiglioma activity. Brain Behav Immun. Mar;85:170-185. Epub 2019/05/07.
  • Johann S, Beyer C. 2013. Neuroprotection by gonadal steroid hormones in acute brain damage requires cooperation with astroglia and microglia. J Steroid Biochem Mol Biol. Sep;137:71-81. Epub 2012/12/01.
  • Maurya SK, Bhattacharya N, Mishra S, Bhattacharya A, Banerjee P, Senapati S, Mishra R. 2021. Microglia Specific Drug Targeting Using Natural Products for the Regulation of Redox Imbalance in Neurodegeneration. Front Pharmacol.12:654489. Epub 2021/05/01.
  • Park J, Lee C, Kim YT. 2023. Effects of Natural Product-Derived Compounds on Inflammatory Pain via Regulation of Microglial Activation. Pharmaceuticals (Basel). Jun 29;16. Epub 2023/07/29.
  • Wen R, Lv J, Jia P, Yang W, Wang N, Wu X, Xue Z, Liu Y. 2022. The protective effects of natural product tunicatachalcone against neuroinflammation via targeting RIPK2 in microglia BV-2 cells stimulated by LPS. Bioorganic & medicinal chemistry. Sep 1;69:116916. Epub 2022/07/07.
  • Xie Q, Wu GZ, Yang N, Shen YH, Tang J, Zhang WD. 2018. Delavatine A, an unusual isoquinoline alkaloid exerts anti-inflammation on LPS-induced proinflammatory cytokines production by suppressing NF-kappaB activation in BV-2 microglia. Biochem Biophys Res Commun. Jul 12;502:202-208. Epub 2018/05/25.

Author Response

1. According to the Manuscript Title and Introduction (Lines 50–51), a connection between NPs and the treatment of neuroinflammation at the microglial cell level should be focused, clarified, and explained in detail. However, the authors have overlooked numerous citations and an in-depth discussion on how microglial cells modulate the neuroinflammation through the effects of various NPs (e.g., alkaloids, flavonoids, etc.) on their membranes or specific targets, as illustrated in Figure 2. Examples of relevant citations are provided in the reference list below, and the authors are encouraged to include more references to strengthen this perspective. Additionally, the authors should compile this information into a detailed new table with the following column headings:

    • Natural product (Extract or Compound name)
    • Mechanism of Action (Linked to microglial cell targets)
    • Model used (e.g., In vitro, In vivo, molecular modeling, with or without molecular dynamics simulations [MDS], etc.)
    • Cited references

Answer: Thank you for your valuable comments. In accordance with your suggestion, I have revised the manuscript by incorporating examples of different types of natural products that inhibit neuroinflammation.

2. Based on the expanded discussion and the new table, as suggested above, Figure 2 should be revised to reflect the updated information.

Answer: Thank you for your valuable comments. However, I would like to retain Figure 2 in its original style. I attempted to add the compound names to Figure 2, but it compromised the clarity and made the figure appear cluttered.

3. Detailed information on active compounds, e.g., those mentioned in Line 343: "In vitro assays identified multiple 342 active compounds (4–17, 19, 20, 22, 23, 26, 29, and 31), with compound 6 showing..." are required. Each compound should be individually named as mentioned in literature, its chemical nature, along with the corresponding reference cited. Additionally, the chemical identity/name of "compound 6" and other numbered compounds throughout the manuscript must be explicitly provided.

Answer: Thank you for your meaningful comments. According to your suggestion, I added compounds name on manuscript.

4. All abbreviations must be defined upon their first appearance in the text. Examples include RelA, IκB, c-Rel, TREM2, PKa, etc., which currently lack full explanations for the reader.

Answer: Thank you for your valuable comments. I have reviewed the abbreviations used in the manuscript and have added their full names where applicable. However, "Rel" is a protein in the NF-κB signaling pathway that does not have a full name; therefore, I was unable to provide one.

5. For Table 1, which lists NPs treating neurological diseases (irrespective of their effects on microglial cells), additional columns should be added for: cited references and clinical trial phase (I, II, III, in market).

Answer: Thank you for your valuable comments. I have included information on the clinical trial phases and added the corresponding trial numbers to facilitate future readers in easily searching for trial details online.

6. Latin names of plants or organisms should be italicized throughout the manuscript for consistency and accuracy.

Answer: Thank you for your meaningful comments. I revised the manuscript.

Round 2

Reviewer 2 Report

Comments and Suggestions for Authors

The authors have addressed all the concerns, and there is no comment any more.

Reviewer 3 Report

Comments and Suggestions for Authors

None